# Phase codes emerge in recurrent neural networks optimized for modular arithmetic

**Keith T. Murray**
Princeton Neuroscience Institute
Princeton University
Princeton, NJ 08544
km3199@princeton.edu

## Abstract

Recurrent neural networks (RNNs) can implement complex computations by leveraging a range of dynamics, such as oscillations, attractors, and transient trajectories. A growing body of work has highlighted the emergence of phase codes, a type of oscillatory activity where information is encoded in the relative phase of network activity, in RNNs trained for working memory tasks. However, these studies rely on architectural constraints or regularization schemes that explicitly promote oscillatory solutions. Here, we investigate whether phase coding can emerge purely from task optimization by training continuous-time RNNs to perform a simple modular arithmetic task without oscillatory-promoting biases. We find that in the absence of such biases, RNNs can learn phase code solutions. Surprisingly, we also uncover a rich diversity of alternative solutions that solve our modular arithmetic task via qualitatively distinct dynamics and dynamical mechanisms. We map the solution space for our task and show that the phase code solution occupies a distinct region. These results suggest that phase coding can be a natural but not inevitable outcome of training RNNs on modular arithmetic, and highlight the diversity of solutions RNNs can learn to solve simple tasks.

## 1   Introduction

Recurrent neural networks (RNNs) are widely used in computational neuroscience to model the computations and dynamics of biological circuits [1–3]. Strikingly, the dynamics and representations learned by task-optimized RNNs often resemble those found in biological systems, despite clear differences in their implementation details [4–7]. One tempting explanation for this convergence is that networks learn these solutions because they are uniquely optimal for performing the task [8, 9]. However, artificial networks are optimized not only for task performance but also under a variety of inductive biases encoded in their architectures, loss functions, or training procedures. These design choices raise the question of whether networks discover certain solutions because they are truly task-optimal, or because those solutions reflect a joint optimization of the task and the inductive biases imposed during training.

A notable example of this convergence between artificial and biological networks is in phase codes, a neural mechanism where information is represented in the relative timing of oscillatory activity. Such codes have been widely observed in the brain [10–12] and have been reproduced in RNNs optimized for working memory tasks [13–16]. However, these prior RNN studies incorporated inductive biases that explicitly favor oscillatory solutions. For instance, Pals et al. [13] embedded oscillatory structure directly into the task objective by training RNNs to output a desired oscillation. Duecker et al. [14], Effenberger et al. [16] constrained hidden unit dynamics to obey second-order differential equations, effectively modeling each unit as a driven harmonic oscillator. Lastly, Liebe et al. [15] introduced a spectral regularization term that explicitly increased power at selected

frequency components of the RNN's computed local field potential (LFP). Hence, it remains unclear whether RNNs learned phase codes because they are genuinely optimal for task performance, or simply because they were biased to learn them.

Here, we address this question by training continuous-time RNNs on a simple modular arithmetic task while deliberately avoiding previous inductive biases that encourage oscillations. Beyond asking whether phase codes emerge, we explored the space of solutions learned by networks across hyperparameter settings and random initializations. This allowed us to characterize a variety of distinct dynamical mechanisms that all solve our task. We find that phase code solutions do arise in a subset of trained networks but are not universal. By mapping this solution space with Dynamical Similarity Analysis (DSA) [17], we show that phase codes are a natural but not inevitable outcome of task optimization.

## 2   Task

We designed a modular arithmetic task to which we refer as the modulo-3 arithmetic (M3A) task. On each trial, the network received a sequence of three discrete inputs (integers 0, 1, or 2), each presented as a scaled[1] one-hot encoded vector with amplitude 5. The network was tasked to indicate whether the sum of the three presented integers was congruent to $0$ modulo 3, outputting $+1$ if congruent and $-1$ otherwise. We refer to trials where the sum is congruent to 0 modulo 3 as *congruent trials*, and trials where the sum is incongruent as *incongruent trials*. Figure 1 depicts the setup of the M3A task.

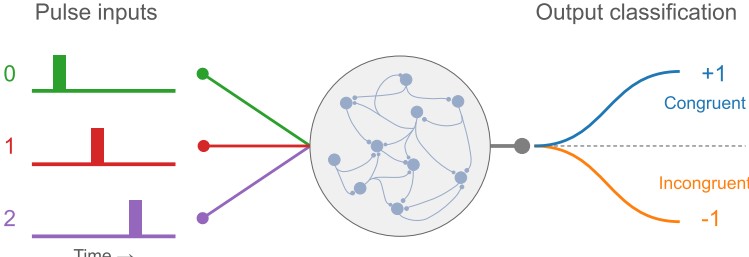

Figure 1: **Schematic of the modulo-3 arithmetic (M3A) task.** The network receives inputs through three distinct channels, each representing one of the integers 0, 1, or 2. On each trial, three integers are presented sequentially as brief pulses. The network's task is to output $+1$ if the sum of the integers is congruent to 0 modulo 3, and $-1$ otherwise.

Trials unfolded over a 1-second period, discretized into $T = 50$ timesteps with $\Delta t = 20\,\mathrm{ms}$. For each trial, three input integers were uniformly sampled from $\{0, 1, 2\}$. The input presentation timesteps were independently drawn from a uniform distribution $t \sim U[6, 45]$, with each input pulse lasting two consecutive timesteps (40 ms). Input presentations were spaced at least five timesteps apart (minimum inter-pulse interval of 100 ms), with presentation times resampled if this spacing constraint was not met. Datasets consisted of 2500 training trials, 900 validation trials, and 540 testing trials, with each dataset balanced between congruent and incongruent trials.

## 3   Results

We trained continuous-time RNNs (see A.1) to perform the M3A task without architectural or regularization constraints that explicitly promote oscillatory dynamics (see A.2). Across training runs, we observed a range of solutions with phase codes emerging in a subset of networks.

### 3.1   Phase code solution

A subset of trained networks learned limit cycle solutions in which integer pulses were encoded as phase shifts along the cycle (for training details, see B.1 and B.2). Each integer pulse advanced

---

[1]Scaling one-hot encoded integer vectors is not necessary to the task but results in quicker training.

or delayed the network's phase such that the final output of the network reflected the cumulative sum of all phase shifts in the trial (Fig. 2a–c). Endpoints of network activity ($t = T$) across all training trials clustered according to the final modular sum, forming three distinct regions in principal component space corresponding to congruence classes 0, 1, and 2 (Fig. 2d). Phase response curves (PRCs; see A.3) showed that each integer pulse induced a consistent phase shift regardless of the network's current state along the limit cycle (Fig. 2e). Summing mean PRC values for each integer sequence produced clusters that mapped onto congruence classes 0, 1, and 2 (Fig. 2f), mirroring the endpoint clusters of network activity observed in principal component space (Fig. 2d).

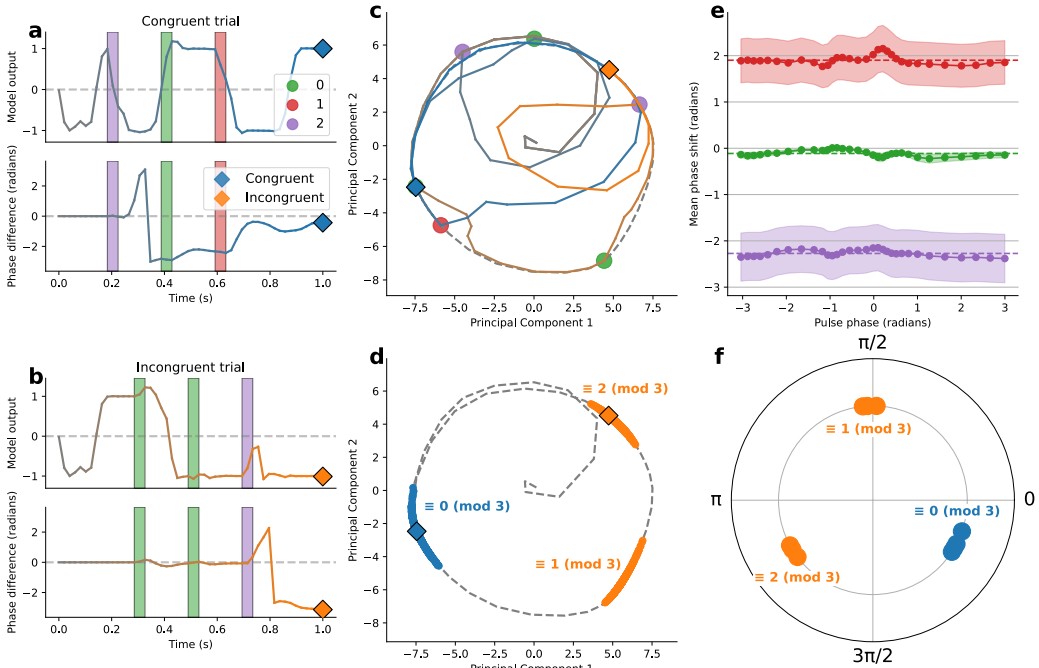

Figure 2: **Phase code solution in an RNN trained on the M3A task. (a)** Network output for a congruent trial (top row) and the corresponding phase difference relative to null activity (bottom row). The network outputs $+1$ at the end of the trial, correctly indicating a congruent input. The phase difference responds to integer pulses, settling near zero by trial end. **(b)** Same as panel (a), but for an incongruent trial. **(c)** Network activity projected onto the first two principal components for the example trials in panels (a) and (b). Trajectories evolve along a limit cycle, with integer pulses inducing phase shifts (advances or delays). Null activity is shown as a dashed gray line. **(d)** Endpoints ($t = T$) of testing trials in PCA space cluster along the null activity in three distinct regions. Each cluster corresponds to a final sum congruent to 0, 1, or 2 modulo 3. **(e)** Phase response curves (PRCs) for integer pulses 0 (green), 1 (red), and 2 (purple). Each PRC shows a consistent shift in phase regardless of when the input is presented. Error bars represent standard deviation. **(f)** Summed mean PRC values for each integer sequence predict summed shifts in network phase. These summed phase values recapitulate the clustering observed in panel (d).

## 3.2 M3A solution space

While phase codes emerged as one viable solution to the M3A task, it was not the only one. To explore the solution space of M3A, we trained 90 RNNs across a range of architectural hyperparameters (see A.1) and applied DSA to compare the dynamics of all trained models (see A.4). A multidimensional scaling (MDS) projection of the resulting DSA embeddings revealed a heterogeneous solution space where the phase code solution from Fig. 2 occupied one distinct region, but was accompanied by a diversity of qualitatively different solutions throughout solution space (Fig. 3a).

Inspection of a select few representative models revealed several qualitatively distinct strategies for solving the M3A task. While additional solutions were observed, we highlight these four due

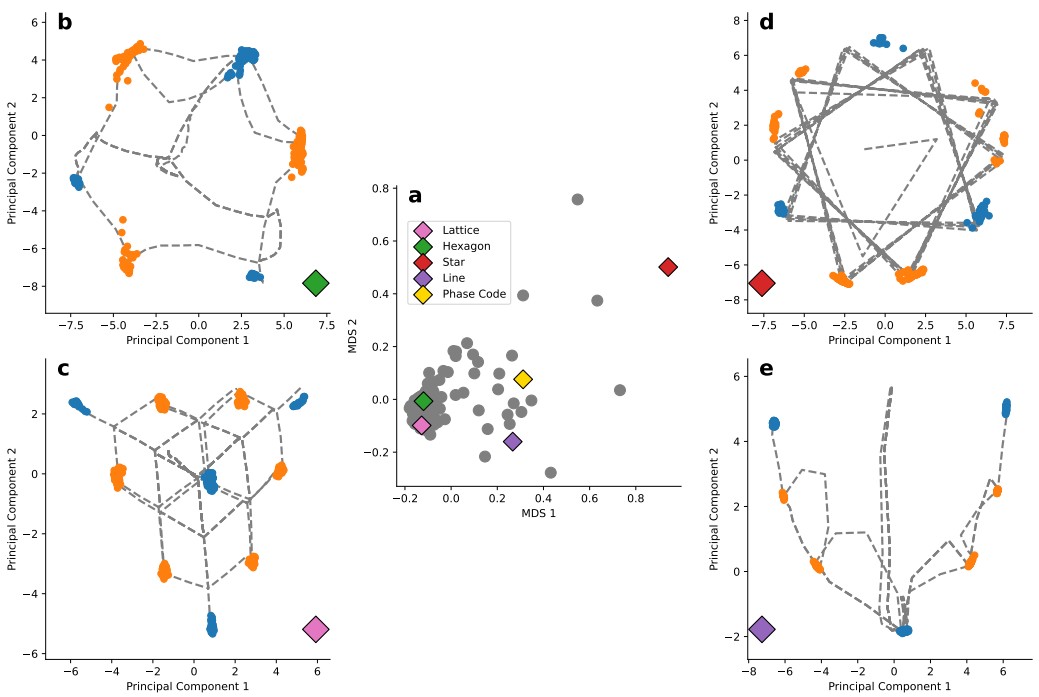

Figure 3: **Heterogeneity in the solution space of RNNs trained on the M3A task.** An architecture hyperparameter search reveals that phase coding is just one of many viable strategies for solving the M3A task. **(a)** MDS projection of Dynamical Similarity Analysis (DSA) embeddings for 91 trained RNNs. Each point represents one trained model. The gold diamond marks the phase code solution from Fig. 2, and colored diamonds highlight models explored in panels (b–e). **(b)** PCA endpoint plot for the *hexagon* solution (green diamond). Trial endpoints converge to fixed-point attractors arranged in a hexagon pattern. Integer pulses transition the network between these attractors in a clock-like fashion. **(c)** Same as (b), but for the *lattice* solution (pink diamond). Attractors are arranged in a lattice-like configuration, and integer pulses drive vector-like transitions across the lattice. **(d)** Same as (b), but for the *star* solution (red diamond). The network exhibits fast oscillations that produce a star-like null activity. Unlike the phase code solution (Fig. 2d), endpoints form nine clusters instead of three. **(e)** Same as (b), but for the *line* solution (purple diamond). Attractors lie along a one-dimensional axis and each integer pulse shifts network activity along this axis.

to their clarity and mechanistic diversity. One model implemented a *hexagon* solution in which trial trajectories converged to fixed-point attractors arranged in a hexagon configuration, and integer pulses drove transitions among attractors in a clockwise fashion (Fig. 3b). Another model exhibited a *lattice* solution with attractors positioned in a lattice-like configuration and integer pulses produced vector-like shifts across this lattice (Fig. 3c). A third model implemented a *star* solution characterized by fast oscillations that formed a star-shaped null trajectory with endpoints distributed across nine distinct clusters (Fig. 3d). Lastly, a *line* solution emerged in which attractors were positioned along a one-dimensional axis and integer pulses incrementally drove network activity along this axis (Fig. 3e).

In total, visual inspection revealed that 6 out of the 90 trained RNNs exhibited a phase code solution. All six networks shared two hyperparameters: a tanh activation function and a time constant of $\tau = 20\,\text{ms}$. Notably, only 15 of the 90 RNNs were trained with this configuration (see A.1) with the remaining 9 networks learning solutions characterized by fixed-point dynamics. This suggests that while not sufficient to induce oscillatory dynamics, this hyperparameter setting may predispose networks toward phase code solutions.

# 4 Discussion

Prior work has shown that RNNs can learn phase codes to represent information in working memory tasks, yet these studies incorporated oscillatory biases to promote such phase codes [13–16]. Our findings show that phase codes can emerge naturally in purely task-optimized networks, although not in every training instance. This raises several questions: What specific task properties, inductive biases, or initial conditions promote phase codes? Are there tasks where phase codes are the universally optimal solution? What computational advantages might phase codes confer to elicit a homogeneous solution space? Prior work suggests that oscillatory dynamics can enhance learning efficiency [16], stabilize gradients [18], and segregate competing information [14], advantages that could potentially explain a homogeneous solution space characterized by phase codes.

Our study has three notable limitations. First, the M3A task is intentionally simple, making it easier for a variety of RNN dynamics to perform the task. Previous work has noted that more complex tasks tend to exhibit more homogeneous solution spaces [19], so it is possible that more complex versions of modular arithmetic, such as M4A or M5A, might not admit a heterogeneous solution space or phase code solutions. Second, while we systematically varied hyperparameters, we did not exhaustively explore all potential hyperparameters. A variety of architectural constraints, like gating mechanisms [20], were notably left out and could add another dimension of solution space diversity that was not captured. Third, while we intentionally avoided the explicit architectural and regularization biases used in prior work to promote oscillatory dynamics [13–16], we observed that all RNNs that learned a phase code solution shared the same activation function and time constant. However, this configuration also gave rise to non-oscillatory solutions, suggesting that while these hyperparameters may predispose networks toward oscillatory dynamics, they are not sufficient to induce them. Future work could more systematically probe the role of these and other hyperparameters in shaping the solution space.

We found that M3A admits a heterogeneous range of solutions, in contrast to simpler tasks like the three-bit flip-flop (3BFF) or context-dependent integration (CDI) tasks which have been shown to admit more homogeneous solution spaces [8, 21, 22]. In line with our findings, previous work has found RNNs to have diverse solution spaces for simple tasks [23]. Therefore, it remains an open question as to why some tasks, like M3A, support a heterogeneous solution space, while others, like 3BFF and CDI, support more homogeneous solution spaces. Altogether, our findings highlight the richness of RNN solution spaces and underscore the need to treat emergent solutions as contingent outcomes of optimization, rather than inevitable products of a task's structure alone.

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

# A  Methods

All code was written in Python using the JAX and Flax packages [24]. Our code is publicly available at `https://github.com/keith-murray/emergence-phase-codes`.

## A.1  Model Architectures

We trained continuous-time recurrent neural networks (RNNs) described by the differential equation

$$\tau \dot{x}_i(t) = -x_i(t) + \sum_{k=1}^{N} W_{ik}^{\text{rec}} f\big(x_k(t)\big) + \sum_{k=1}^{N^{\text{in}}} W_{ik}^{\text{in}} u_k(t) + b_i^{\text{rec}} + \eta_i(t) \tag{1}$$

where $x_i(t)$ denotes the membrane voltage (or activity) of recurrent unit $i$, $\tau$ is the network time constant, and $u_k(t)$ represents the task input. We used $N = 100$ recurrent units and $N^{\text{in}} = 3$ input channels corresponding to the integers $\{0, 1, 2\}$. The function $f(\cdot)$ is a pointwise nonlinearity that maps voltage to firing rate. $\mathbf{W}^{\text{rec}}$ and $\mathbf{W}^{\text{in}}$ denote the recurrent and input weight matrices, respectively, and $b^{\text{rec}}$ is a bias term. At each time step $t$, $\eta_i(t)$ is Gaussian noise drawn from $\mathcal{N}(0, \sigma^2)$ and added to the recurrent unit activity. Networks were discretized via Euler's method with a time step of $\Delta t = 20\,\text{ms}$. The network output was computed as a linear readout of the firing rates

$$y(t) = \sum_{k=1}^{N} W_{1,k}^{\text{out}} f\big(x_k(t)\big) + b^{\text{out}} \tag{2}$$

where $y(t)$ is the output unit, $\mathbf{W}^{\text{out}}$ is the readout weight matrix, and $b^{\text{out}}$ is a bias term. The weights $\mathbf{W}^{\text{rec}}$, $\mathbf{W}^{\text{in}}$, and $\mathbf{W}^{\text{out}}$ were initialized using Glorot normal initialization [25] and biases $b_i^{\text{rec}}$ and $b^{\text{out}}$ were initialized to zero. Hidden states $x_i(0)$ were set to 1 at the beginning of each trial.

To map the space of possible solutions for RNNs trained on the M3A task, we trained models for every combination of the following parameters: activation function $f \in \{\tanh, \text{ReLU}\}$, time constant $\tau \in \{20\,\text{ms}, 40\,\text{ms}, 200\,\text{ms}\}$, and recurrent noise standard deviation $\sigma \in \{0.00, 0.05, 0.10\}$. For each parameter combination, we trained RNNs with five different random seeds, yielding $2 \times 3 \times 3 \times 5 = 90$ total networks.

## A.2  Task Optimization

Continuous-time RNNs were optimized on the M3A task according to

$$E_{\text{task}} = \frac{1}{MT_{\text{train}}} \sum_{m=1}^{M} \sum_{t=T-2}^{T} \big(y(t, m) - y^{\text{target}}(t, m)\big)^2 \tag{3}$$

where $y^{\text{target}}(t, m)$ denotes the target output signal indicating congruence (+1) or incongruence (–1). Only the last three timesteps $T_{\text{train}} = 3$ of each trial were included in (3). RNNs were trained on batches of $M = 16$ trials. We included a metabolic penalty on firing rates given by

$$R_{\text{rates}} = \frac{1}{MTN} \sum_{m,t,i=1}^{M,T,N} f\big(x_i(t, m)\big)^2 \tag{4}$$

which encourages sparse or low-energy activity [5]. The complete loss function was

$$\mathcal{L} = E_{\text{task}} + \alpha R_{\text{rates}} \tag{5}$$

where $\alpha$ is a hyperparameter controlling the strength of the regularization, set to $\alpha = 10^{-4}$ in all experiments. Note that the inclusion of a metabolic penalty on firing rates does not contradict our goal of avoiding oscillatory-promoting biases, as prior work has shown that RNNs trained with such penalties can still exhibit transient [26] and fixed-point dynamics [27].

All network parameters ($\mathbf{W}^{\text{rec}}$, $\mathbf{W}^{\text{in}}$, $\mathbf{W}^{\text{out}}$, and corresponding biases) were optimized according to (5) using the AdamW optimizer [28], with parameters $\lambda = 10^{-4}$, $\beta_1 = 0.9$, $\beta_2 = 0.999$, and $\epsilon = 10^{-8}$. RNNs were trained for up to 1000 epochs, with early stopping applied if the validation loss (5) fell below 0.001.

### A.3 Phase response curve

A phase response curve (PRC) is a method for assessing the impact a presented stimulus has on the phase of an oscillator [29]. To estimate PRCs for an RNN, we first measured the network's oscillatory period by identifying peaks in a one-dimensional projection of the network activity, defined by the first principal component of firing rates. After estimating the period, we systematically applied brief perturbing input pulses at all phases of the oscillation period. For each input type (corresponding to integers 0, 1, and 2), we injected a scaled one-hot pulse at a given phase and recorded the resulting network activity over one period. The instantaneous phase was computed via the arctangent of the first two principal components, and the phase shift was defined as the difference between the perturbed and unperturbed phases (wrapped to $[-\pi, \pi]$). Repeating this procedure across all input types and pulse timings yielded a three-dimensional tensor of shape $(3, \text{period}, \text{period})$, capturing the phase shifts induced by each input across the entire oscillation period. PRCs were visualized and summarized by averaging phase shifts across the measured oscillation period to estimate the magnitude of phase resets.

### A.4 Assessing model similarity

We used Dynamical Similarity Analysis (DSA) to pairwise compare the learned dynamics across trained RNNs. We chose DSA because it has been shown to assess network topological structure while being invariant to individual differences in representational geometry and residual dynamics [17]. For each trained RNN, we first estimated a linear operator $A$ that maps activity from timestep $t$ to timestep $t+1$. To compare two RNNs $X$ and $Y$, DSA measures the distance between the RNNs' linear forward operators $A_X$ and $A_Y$ by minimizing

$$d_{\text{DSA}}(A_X, A_Y) = \min_{C \in O(n)} \left\| A_X - C A_Y C^{-1} \right\|_F \qquad (6)$$

where $O(n)$ denotes the orthogonal group and $\| \cdot \|_F$ denotes the Frobenius norm. Pairwise DSA distances were assembled into a distance matrix, which we then embedded in two dimensions using multidimensional scaling for visualization (Fig. 3a).

## B Phase code solution

### B.1 RNN parameters

The continuous-time RNN parameters (see A.1) used to generate the phase code solution in Fig. 2 were as follows: activation function $f = \tanh$, time constant $\tau = 20\,\text{ms}$, and recurrent noise standard deviation $\sigma = 0.05$. One potential concern is that these parameter choices might bias the network toward learning oscillatory dynamics. However, the *line* solution shown in Fig. 3e was trained using the same parameters, yet it converged to fixed-point dynamics. This suggests that these parameter choices are not sufficient to impose an oscillatory bias.

### B.2 Training dynamics

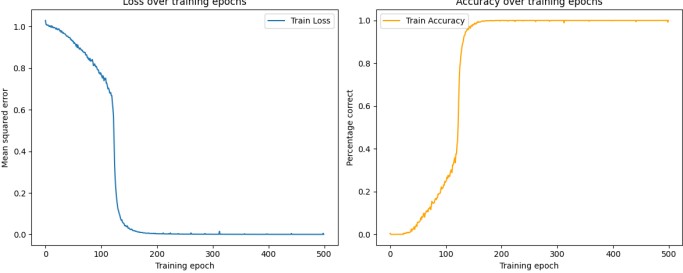

Figure 4: **Training dynamics of the phase code solution. Left:** Training loss (mean squared error) over 500 epochs. **Right:** Percentage of correct classifications during training. The network converges rapidly to near-zero error and near-perfect accuracy after approximately 150 epochs.

