# OpenReview forum: "Phase codes emerge in recurrent neural networks optimized for modular arithmetic"
_NeurIPS.cc/2025/Workshop/UniReps — UniReps2025_

### Official Review · Reviewer_bcwq · 2025-09-10
**In this work, the authors present exploratory findings about how purely task-driven objectives contribute to the structure of embeddings for continuous-time RNNs. Their paper provides interesting insights, interpretations, and inspirations for future work**

**Confidence:** 4

**Review:**

In this work, the authors present exploratory findings about how purely task-driven objectives contribute to the structure of embeddings for continuous-time RNNs. Specifically, the authors explore the task of Modulo-3 Arithmetic with pulse inputs and congruent/incongruent output targets. A certain temporal sequence of pulses from 3 channels must be interpreted via modulo arithmetic into a -1 or +1 output.

Such modulo operations tend to encourage oscillatory solutions, and the authors wanted to explore whether a trained network would discover an oscillatory solution space without inductive bias–in this work, they only provided a task-driven regression objective, without the biases of previous works.

With their presented training framework, the authors found that, yes, the embedding space of trained (and minimally biased) RNNs did sometimes produce oscillatory limit cycle solutions, as well as a spectrum of other arrangements to the embedding space.

These results are fascinating and provide an interesting starting point for future explorations.

Open-Ended Questions:
- Why does PCA project the embedding space to a circle? Does that indicate that the dominant dimension relates to the phase cycle? Do the first two dimensions of PCA dominate? Could this task be solved with just one PCA dimension–or are two dimensions required?
- The plots are very cool, but it seems like PCA is doing a fair amount of heavy lifting. PCA is a projection, so it would be interesting to provide some insights about the structure in the higher dimensional space.
- The plot showing the layout of different solution modes and their relative similarity was interesting. It seems like there’s a large cluster in the Lattice/Hexagon area, and Phase Codes stand apart.
- Are these results particularly surprising? I do think it is interesting that a discrete set of PCA patterns emerge from the same-task driven objective. Maybe the more surprising aspect is that, for this essentially binary classification problem, we see more than just two clusters in embedding space. I wonder why we see more than two clusters, and moreover, if there is any significance to the number of clusters.

Quality:
This paper is high quality, with concise writing and interesting insights.

Clarity:
This paper is clear, with helpful visualizations and appropriate references to the appendix. The introduction could have included some concrete examples from the cited works, to make the prose more self-contained.

Originality:
The paper seemed to present a novel finding at the intersection of two spaces–continuous-time RNNs and structured embedding spaces. There is likely originality at this intersection. It would be helpful to include more references to how other fields have analyzed the structure of embedding spaces and interpretability of PCA.

Significance:
This paper is a solid exploratory step into analyzing emergent, task-driven structure to continuous-time RNNs. I think it will inspire interesting follow-up studies.

**Score:**

4

**Topic Fit:**

3

---

### Official Review · Reviewer_sQZ9 · 2025-09-14
**Interesting findings but needs more rigorous analysis**

**Confidence:** 4

**Review:**

Quality: Reasonable setup but lacks rigor. Missing key details like number of random seeds, statistical testing, and robustness checks. The solution categorization seems subjective.
Clarity: Well-written with clear figures. Figure 2 shows phase precession nicely, Figure 3 illustrates solution diversity well. Some concepts need better definition though.
Originality: Limited novelty. That RNNs find multiple solutions is known. Spontaneous phase coding is somewhat interesting but not groundbreaking.
Significance: Hard to assess given the toy task. Modulo-3 arithmetic is too simple, unclear if findings generalize to real cognitive functions.

Pros: Clear research question about spontaneous oscillations, good visualization of solution diversity, proper controls without oscillatory biases, and accessible writing.

Cons: Missing crucial methodological details, no theoretical insight into why solutions emerge.

**Score:**

4

**Topic Fit:**

2

---

### Official Review · Reviewer_pwTZ · 2025-09-15
**Phase codes emerge in recurrent neural networks optimized for modular arithmetic - Review**

**Confidence:** 3

**Review:**

Thank you for all your nice work unraveling the structure of the solution space of Recurrent Neural Networks!

The paper aims to study the emergence of phase codes, related to oscillatory solutions, on Recurrent Neural Networks modelling a simple modular arithmetic task. The hypothesis they want to study is whether the existence and predominance of oscillatory responses arise naturally because of optimality for the task, or are influenced by the architectural and hyperparameter choices. The idea is very innovative and yet still underexplored.

There are some minor issues, however, that should be addressed. All issues are presented below:

Introduction

Throughout the paper, the authors mention the existence of multiple inductive biases imposed by the architectures that encourage the emergence of oscillatory dynamics (lines 24, 32). What would be these explicit factors, and how do the configurations explored eliminate them?

It is also mentioned (line 31) that similar phase codes have been reported in RNNs optimized for working memory tasks. Since this paper investigates a different task (modular arithmetics), to what extent can we extend the findings of the paper and conclude whether oscillatory dynamics in memory tasks are genuinely task-optimal?

Results

The paper discusses the multiple phase codes and non-phase codes solutions for the task. It mentions the presence of a diverse set of distinct non-phase code solutions. How many of the trained networks ended up in the phase-code category vs other categories?

Were there any identifiable hyperparameters that predisposed the network to settle into a phase code or a non-phase code solution?

**Score:**

4

**Topic Fit:**

2